# Radiation-Induced Metabolic Shifts in the Hepatic Parenchyma: Findings from ^18^F-FDG PET Imaging and Tissue NMR Metabolomics in a Mouse Model for Hepatocellular Carcinoma

**DOI:** 10.3390/molecules26092573

**Published:** 2021-04-28

**Authors:** Yi-Hsiu Chung, Cheng-Kun Tsai, Ching-Fang Yu, Wan-Ling Wang, Chung-Lin Yang, Ji-Hong Hong, Tzu-Chen Yen, Fang-Hsin Chen, Gigin Lin

**Affiliations:** 1Department of Medical Research and Development, Linkou Chang Gung Memorial Hospital, Taoyuan 333, Taiwan; stella720905@gmail.com; 2Clinical Metabolomics Core Lab, Chang Gung Memorial Hospital at Linkou, Taoyuan 333, Taiwan; klem.tsaick@gmail.com; 3Radiation Biology Research Center, Institute for Radiological Research, Chang Gung University/Chang Gung Memorial Hospital, Linkou, Taoyuan 333, Taiwan; chingfang@mail.cgu.edu.tw (C.-F.Y.); jihong@adm.cgmh.org.tw (J.-H.H.); 4Department of Radiation Oncology, Chang Gung Memorial Hospital-LinKou, Taoyuan 333, Taiwan; 5Department of Nuclear Medicine, Chang Gung Memorial Hospital, Taoyuan 333, Taiwan; sweet216014@hotmail.com (W.-L.W.); yentc1110@gmail.com (T.-C.Y.); 6Department of Medical Imaging and Radiological Sciences, Chang Gung University, Taoyuan 333, Taiwan; a0955498306@gmail.com; 7Department of Medical Imaging and Intervention and Institute for Radiological Research, Chang Gung Memorial Hospital at Linkou and Chang Gung University, Taoyuan 333, Taiwan

**Keywords:** glycolysis, liver cancer, radiation, 18F-FDG PET, NMR

## Abstract

Purpose: By taking advantage of 18F-FDG PET imaging and tissue nuclear magnetic resonance (NMR) metabolomics, we examined the dynamic metabolic alterations induced by liver irradiation in a mouse model for hepatocellular carcinoma (HCC). Methods: After orthotopic implantation with the mouse liver cancer BNL cells in the right hepatic lobe, animals were divided into two experimental groups. The first received irradiation (RT) at 15 Gy, while the second (no-RT) did not. Intergroup comparisons over time were performed, in terms of 18F-FDG PET findings, NMR metabolomics results, and the expression of genes involved in inflammation and glucose metabolism. Results: As of day one post-irradiation, mice in the RT group showed an increased 18F-FDG uptake in the right liver parenchyma compared with the no-RT group. However, the difference reached statistical significance only on the third post-irradiation day. NMR metabolomics revealed that glucose concentrations peaked on day one post-irradiation both, in the right and left lobes—the latter reflecting a bystander effect. Increased pyruvate and glutamate levels were also evident in the right liver on the third post-irradiation day. The expression levels of the glucose-6-phosphatase (G6PC) and fructose-1, 6-bisphosphatase 1 (FBP1) genes were down-regulated on the first and third post-irradiation days, respectively. Therefore, liver irradiation was associated with a metabolic shift from an impaired gluconeogenesis to an enhanced glycolysis from the first to the third post-irradiation day. Conclusion: Radiation-induced metabolic alterations in the liver parenchyma occur as early as the first post-irradiation day and show dynamic changes over time.

## 1. Introduction

Hepatocellular carcinoma (HCC) is the sixth most common cancer worldwide [1,2]. While radiotherapy is an integral part of current HCC treatment protocols, radiation-induced liver disease (RILD) continues to represent a major obstacle to its widespread implementation [3,4]. The onset of RILD is clinically characterized by anicteric hepatomegaly, ascites, and elevated serum alkaline phosphatase. Conversely, atypical signs, include jaundice, as well as elevated transaminase levels, including aspartate aminotransferase (AST) and alanine aminotransferase (ALT) [5]. The onset of RILD in humans generally occurs three-six months after liver irradiation. Whereas, it traditionally appears at two post-irradiation weeks in rodent models [5,6]. Recently, molecular changes in irradiated tissues have shown to precede overt morphological or physiological alternations [7,8]. An in-depth knowledge of early metabolic changes during irradiation might link to the underlying pathophysiology of RILD.

Positron emission tomography (PET) with 2-deoxy-2-[18F] fluoro-D-glucose (18F-FDG) has several key applications in the field of oncology—including diagnosis, tumor staging, and assessment of treatment response [9]. The basic principle underlying this technique lies in its ability to measure glucose uptake [10,11,12]. However, areas of infection or active inflammation may lead to false positive results in up to 13% of all cases [13]. It can be speculated that the increased 18F-FDG uptake in patients with HCC who develop RILD may at least in part stem from inflammatory and/or metabolic mechanisms. Tissue nuclear magnetic resonance (NMR) metabolomics is increasingly being applied as an analytical platform to identify and quantify metabolites under different biological conditions [14,15]. Therefore, we reasoned that this technique would allow extensive analysis of glucose metabolic pathways in irradiated liver samples—ultimately improving our understanding of RILD pathophysiology.

By taking advantage of 18F-FDG PET imaging and tissue NMR metabolomics, we therefore designed the current study to investigate the metabolic alterations associated with liver irradiation in a mouse model for HCC.

## 2. Results

### 2.1. 18F-FDG Uptake in Irradiated and Not-Irradiated Liver Parenchyma

18F-FDG PET/CT scans were conducted on post-RT days 1 and 3 to monitor the 18F-FDG uptake in the liver parenchyma of mice bearing experimental HCC with 5 mm of tumor size. Animals in the no-RT group served as controls. Representative transaxial planes of liver CT and 18F-FDG PET/CT images obtained in the two experimental groups are shown in Figure 1a. On post-RT day 1, a mildly increased 18F-FDG uptake was observed in the right liver parenchyma of irradiated mice. On post-RT day 3, the 18F-FDG uptake in the right liver parenchyma was significantly higher in the irradiated group compared with the non-irradiated group (SUVmax: 1.06 ± 0.29 versus 0.66 ± 0.08, respectively, *p* < 0.01; SUVmean: 0.71 ± 0.13 versus 0.49 ± 0.04, respectively, *p* < 0.05; Figure 1b). However, there was no significant increased in 18F-FDG uptake in the left liver parenchyma in neither RT group nor no-RT group following irradiation (Appendix A). In experiments conducted in tumor-free animals, the right-to-left ratio of 18F-FDG uptake in the liver parenchyma measured on post-RT day 3 was 1.13-fold higher in the RT group compared with the no-RT group (1.11 ± 0.10 versus 0.98 ± 0.03, respectively, *p* < 0.05, Appendix A).

### 2.2. Metabolic Changes in Irradiated and Not-Irradiated Liver Parenchyma

On post-RT day 3, PCA plots revealed significant differences with respect to metabolite concentrations in the right liver of mice in the RT and no-RT groups (Figure 2), which were not evident for the left lobe. In addition, no significant differences were observed between the RT and no-RT groups in either lobe on post-RT day 1. Changes in the expression of each metabolite in the RT and no-RT groups are summarized in Table 1. On post-RT day 1, significant increases in the following metabolites were observed in the left liver lobe: Alanine, anserine, galactarate, galactitol, glucose, glycylproline, malonate, N-methylhydantoin, and succinate (*p* < 0.05). As for the right liver lobe, significant increases on post-RT day 1 were evident for galactarate, galactitol, glucose, and sucrose (*p* < 0.05). Among different metabolites, the highest elevation was observed for glucose (18- and 17-fold in the left and right liver, respectively). On post-RT day 3, the only metabolite found to be significantly increased in the left liver lobe was fumarate. Other metabolites were decreased, albeit not significantly so. While, significant elevations of glutamate, pyruvate, and sucrose were observed in the right liver, other metabolites were significantly decreased.

### 2.3. Markers of Inflammation and Gluconeogenesis in Irradiated and Not-Irradiated Liver Parenchyma

We have previously shown that local irradiation induces a continuous influx of macrophages at sites in orthotopic hepatic neoplasms [16], which was in turn, associated with a higher tumor 18F-FDG uptake. Herein, we have shown that 18F-FDG uptake is increased in normal hepatic tissue on post-RT day 3. Nonetheless, there was no significant expansion of CD8+ T cells as well as of the F4/80+ or CD68+ macrophage populations in liver tissues on post-RT days 1 and 3 (Appendix A). Taken together, these results indicate that the radiation-induced increase in 18F-FDG uptake in the liver is not related to an enhanced infiltration of immune cells. Analyses of pro-inflammatory cytokines revealed a mild increase in IL-18 expression and a significant increase in IL-6 expression in the right liver of irradiated mice on post-RT day 3 (Figure 3, *p* < 0.05). Differences were also observed with respect of both IL-1*β* and HIF-1*α* in the right liver of irradiated mice on post-RT days 1 and 3. In the left lobe of irradiated mice, we found an overexpression of IL-18 and IL-6 on post-RT day 1 and of HIF-1*α*, IL-1*β*, and IL-6 on post-RT day 3. We speculate that this phenomenon may stem from a bystander effect as only the right portion of the liver was directly irradiated (Appendix A). The results of the qPCR conducted in extracts from right liver tissues revealed a reduction of G6PC gene in irradiated tissue compared with non-irradiated tissue on post-RT day 1. Similarly, expression levels of the FPB1 gene were significantly lower in irradiated tissue compared with non-irradiated tissue on post-RT day 3 (Figure 4).

### 2.4. Assessment of Liver Enzymes in Irradiated Mice Bearing Orthotopic HCC

The extent of liver damage was assessed by measuring serum levels of AST, ALT, and albumin. Compared with control animals, the presence of experimental tumors resulted in increased ALT (36.4 ± 11.00 versus 52.67 ± 8.08 IU/L, respectively, *p* = 0.021) and albumin levels (1.36 ± 0.50 g/dL versus 3.27 ± 0.15 g/dL, respectively, *p* < 0.0001). However, ALT, AST and albumin levels did not show significant differences in the RT and no-RT groups neither on post-RT day 1 nor day 3 (Appendix A). These results indicate that irradiation per se does not increase AST and albumin levels.

### 2.5. Alterations in Hepatic Metabolic Pathways in Response to Irradiation

In the right portion of the liver, the following metabolic pathways were found to be altered on post-RT day 1: Starch and sucrose metabolism, galactose metabolism, and biosynthesis of neomycin, kanamycin, and gentamicin. On post-RT day 3, increased pyruvate and glutamate levels were associated with alterations in the metabolism of several amino acids—including D-glutamine-D-glutamate metabolism, alanine-aspartic acid-glutamic acid metabolism, glycine-serine-threonine metabolism, and arginine-proline metabolism. The metabolic changes predicted by the platform, used in this study (available from the www.metaboanalyst.ca, accessed date 26 November 2021) for the right and left lobes of the liver on post-RT days 1 and 3 are shown in Appendix A. By taking into account the results of 18F-FDG PET/CT imaging, NMR metabolomics, and qPCR, we formulated a theoretical model for radiation-induced metabolic alterations in the right liver on post-RT days 1 and 3, according to which gluconeogenesis and glycolysis were alternatively affected (Figure 5).

## 3. Discussion

Using an animal model, herein, we showed that liver irradiation results in a precise temporal sequence of metabolic alterations in the hepatic parenchyma—which was characterized by an alternate pattern (e.g., early inhibition of gluconeogenesis followed by a switch to glycolysis). We also demonstrated that 18F-FDG PET/CT imaging may serve as a useful surrogate tool for monitoring the occurrence and temporal course of metabolic changes elicited by hepatic irradiation. Contrary to 18F-FDG PET/CT, a common understanding of the mechanism of hepatocyte damage, including AST, ALT, and albumin—perhaps did not serve as useful indicators for 15 Gy radiotherapy induced liver damage.

In the previous report, the increased FDG uptake in HCC tumor lesion was found in post-RT day one to day six due to macrophages infiltration [16]. In this study, we observed that the FDG uptake of the irradiated liver parenchyma was significantly higher than that of non-irradiated group in post-RT day three. We found an increased expression of IL-6, a well-known pro-inflammatory cytokine, in both right and left lobes liver of the RT group on the day three. However, immune cell infiltration did not appreciably increase in the irradiated liver parenchyma, thereby suggesting that immune cell activation was not the major contributor to the increased 18F-FDG avidity in liver tissues in the post-irradiation phase. Caution should be taken to definition of tumor margin for treatment planning using FDG PET because the enhancement of both tissues and lesion FDG uptake may lead to ambiguous tumor margins.

Apart from gluconeogenesis and glycolysis, fatty acid biosynthesis and amino acid metabolism were identified as the most affected metabolic pathways in the liver parenchyma. All of them were found to be altered both in the irradiated right liver and in the contralateral non-irradiated lobe. Intriguingly, the increased 18F-FDG uptake was accompanied by an increased pyruvate-to-glutamate ratio and a reduced expression of the FBP1 gene in the irradiated liver. Altogether, these results point to an enhanced glycolysis as the metabolic milieu underlying the enhanced 18F-FDG avidity at three days post-irradiation. A previous study reported that the biological damage observed in rats following total body irradiation (8 Gy) peaked after 72 h [17]. In accordance with our research, the authors identified radiation-induced alterations in amino acid metabolism—specifically involving the glycine-serine-threonine and the alanine-aspartate-glutamine pathways [17]. Our results may pave the way for the use of radiation-induced alterations in hepatic metabolism as promising biomarkers for monitoring the occurrence and progression of RILD.

### 3.1. Metabolic Switch to Glycolysis Following Irradiation of the Liver Parenchyma

While hepatic gluconeogenesis leads to the synthesis of glucose, glycolysis is an energy production pathway during which one glucose molecule is split into two pyruvate molecules [18]. Previous animal studies have shown that hyperglycemia and increased glycogen stores in the liver can be observed in the early post-irradiation phase, indicating a key role for gluconeogenesis in early radiation response [19,20]. Another study reported decreased glycolysis in mice subjected to whole liver irradiation (10 Gy) on post-RT day 1 [21]. The increased amount of glucose detected in our NMR metabolomics experiments is in keeping with the published literature and—consistently—hepatic 18F-FDG uptake was not significantly increased on post-RT day 1. The increased amount of glucose in the liver parenchyma may, in turn, inhibit G6PC gene expression, lending further support to the inhibition of gluconeogenesis within the first post-irradiation day. However, hepatic glucose metabolism was found to change dramatically on post-RT day three. By that time, the enhanced hepatic 18F-FDG uptake, the increased detection of pyruvate and the decreased expression of the FBP1 gene concordantly suggested that a switch to glycolysis had occurred. Activation of glycolysis can be reflected by an increased in both up-stream metabolites (i.e., glucose and sucrose) and the final down-stream product (i.e., pyruvate) [22]. Taken together, these results indicate that early and delayed alterations in glucose metabolism merit further investigation and scrutiny as promising biomarkers of RILD [23,24]. It remains to be established whether the manipulation of glucose metabolism before, or immediately after, irradiation may result in a decreased production of reactive oxygen species (ROS) and/or proinflammatory molecules [25,26,27].

### 3.2. Inflammatory Response in Irradiated Liver Parenchyma

Radiation is known to induce a proinflammatory tissue response, and there is evidence that IL-6 and NF-kB are among the key molecular mediators of RILD [28]. Moreover, proinflammatory mechanisms have been advocated to explain the increased18F-FDG avidity observed in liver and lung tissues of mice subjected to experimental irradiations [29]. While we did not observe an increased hepatic infiltration of immune cells on post-RT days 1 and 3, the expression of IL-6 in the right liver increased gradually over time. Notably, IL-6 can stimulate glycolysis within the tumor microenvironment [30] and a disturbed glucose metabolism can elicit proinflammatory effects [31,32,33]. The complex interplay between radiation-induced alterations in glucose metabolism and inflammatory mechanisms should be subject to future research.

### 3.3. Bystander Effects in the Left Liver Lobe

While only the right liver lobe was directly irradiated in our study, alterations affecting the tricarboxylic acid cycle, the biosynthesis of fatty acids, and amino acid metabolism were also observed in the left hepatic lobe on post-RT days one and three. These metabolic changes, which were accompanied by significantly increased expression of IL-18 and IL-6 on day 1 as well as of IL-1*β*, HIF-1*α*, and IL-6 on day three are likely the results of the bystander effect. Radiation-induced ROS production elicits the release of pro-inflammatory cytokines [27]. Previous studies have shown that the bystander effect occurring in hepatoma cells irradiated with α-particles was mediated by ROS through a p53-dependent pathway [34,35]. Another study conducted in a rat model reported the occurrence of a bystander effect in the brain (with altered gene expression and evidence of DNA damage) following irradiation of the liver [36]. The bystander metabolic changes in the left liver lobe observed in the current study may stem from paracrine effects elicited by proinflammatory cytokines released from the irradiated right lobe [37].

### 3.4. Future Research Directions

Radiation-induced ROS generation is deemed to play a critical role in determining liver radiosensitivity. Interestingly, a blunted hepatic pyruvate dehydrogenase complex activity has been associated with a reduced production of ROS [25,38]. Future research should address whether specific manipulation of gluconeogenesis and/or glycolysis might reduce the sensitivity of the liver parenchyma to radiation therapy through a modulation of ROS formation.

## 4. Materials and Methods

### 4.1. Experimental Design

At day 10 after implantation of mouse liver cancer BNL cells in the right lobe of the liver, mice underwent T2-weighted magnetic resonance (MR) imaging to determine tumor size. Subsequently, animals were randomly divided into two experimental groups. The irradiation (RT) group consisted of five mice, which underwent partial irradiation (15 Gy) of the right liver lobe on the day of MR imaging. The no-RT group comprised five mice which were not irradiated. 18F-FDG PET scans were performed on days 1 and 3 post-irradiation. Ex vivo NMR metabolomics experiments were carried out on tumor and normal liver parenchyma samples on days 1 and 3 post-irradiation (*n* = 6 for both groups). Serum ALT and AST levels were measured on a biochemical analyzer in both the RT (on days 1 and 3 post-irradiation) and no-RT groups. Ex vivo immunochemical staining of inflammatory markers was performed on day 3 post-irradiation. The hepatic expression of the following genes was also assessed by qPCR in the post-irradiation phase: (1) Genes involved in glucose metabolism—including phosphoenolpyruvate carboxykinase 1 (PCK1), fructose bisphosphatase 1 (FBP1), glucose-6-phosphatase (G6PC), and pyruvate carboxylase (PC); (2) hypoxia induced factor 1, alpha subunit (HIF-1*α*), and (3) genes involved in inflammation—including interleukin (IL)-18, IL-1*β*, and IL-6. Table 2 summarizes the number of animals and the experimental workflow.

### 4.2. Animal Model and Procedures for Irradiation

Ethical approval for all animal experiments was received from the Institutional Animal Care and Use Committee of the National Tsing Hua University (approval number: 10414) and the Chang Gung Memorial Hospital (approval number: 2016010701). The animal model and the procedures used for irradiation have been previously described in detail [16]. In brief, mouse liver cancer BNL cells (1 × 105 cells suspended in 20 µL of HBSS) were orthotopically implanted with a 30-G needle in the right liver lobe of 8-week-old male Balb/c mice, purchased from National Laboratory Animal Center (Taipei, Taiwan). The animals were housed in a climate-controlled facility on a 12 h light/12 h dark cycle with access to water and food *ad libitum*. To simulate clinical scenario, the orthotopic HCC mouse model was chosen to investigate the side-effect of radiation therapy. Before irradiation, mice were put under general anesthesia with a mixture (1:1) of ketamine (50 mg/mL) and 1% xylazine and restrained using an adhesive tape to minimize the respiration rate and movement. The irradiation was implemented, while the mice were under the stable respiration movement. The previous reports of liver tumor control by stereotactic body radiation therapy were using high dose per fraction per day, with 3–5 fractions. The radiation dose 15 Gy is a clinically relevant daily dose that achieves over 50% tumor control rate without normal tissue toxicity [39]. Therefore, the right liver was partially irradiated with 20 × 10 mm collimated irradiation field and 6-MV X-ray beams (15 Gy) obtained from a clinical linear accelerator and covered with a 0.5-cm bolus to the animals’ skin [16].

### 4.3. 18F-FDG PET/CT Imaging Protocol and Analysis

While, 18F-FDG PET/CT whole-body scans obtained from experimental animals were retrospectively retrieved from the dataset used for our previous study [16]. In brief, anesthetized mice underwent computed tomography (CT) using ExiTronTM nano6000 (0.1 mL) as contrast agent. Within one hour, animal still kept under anesthesia were subjected to 18F-FDG PET imaging (scan duration: 10 min in supine position) after injection of 18F-FDG (8.1 MBq). PET and CT images were acquired on the same animal bed using the InveonTM (Siemens Medical Solutions Inc., Malvern, PA, USA), and the NanoSPECT/CT (Mediso Kft., Budapest, Hungary) systems, respectively. The volumes of interest (VOIs) within the liver parenchyma were manually identified on PET/CT fused images. The iodine-based CT contrast agent- ExiTronTM nano6000 was accumulated within the reticuloendothelial system such as liver and spleen [40], which lead to the negative contrast images on the tumors. The upper portion of the left liver lobe was not included in VOIs identification because of the spillover from myocardium or myocardial motion. As for VOIs definition in the right liver parenchyma, caution was exercised to avoid the inclusion of both the tumor area and extrahepatic regions. The standardized uptake value (SUV) was calculated by multiplying tissue concentration of the tracer by the animal body weight divided by the injected dose. Image analysis was carried out with PMOD version 4.0 (PMOD Technologies Ltd., Zurich, Switzerland).

### 4.4. Collection and Extraction of the Liver Tissue for NMR Metabolomics

Mice in the RT and no-RT groups (*n* = 6 each) were anesthetized as described above before undergoing portal vein perfusion. Samples were removed bilaterally from the liver parenchyma and stored (100 µg each) at −80 °C in CryoTube™ vials (1.8 mL; Thermo Fisher Scientific, Waltham, MA, USA). Aqueous extracts were resuspended in a buffer (650 µL) containing 0.008% trisodium phosphate, 7.5 mM Na_2_HPO_4_, 0.2 mM NaN_3_, and 92% D_2_O. After centrifugation at 4 °C for 5 min, sample supernatants (600 µL) were withdrawn in SampleJet NMR tubes (Bruker, Billerica, MA, USA).

### 4.5. NMR Metabolomics Data Processing

NMR metabolomics analyses were conducted on a Bruker AVANCE III HD System equipped with a 14.1-T magnet running at 600 MHz (1H NMR) and 279 K. NMR spectra were acquired using nuclear Overhauser enhancement spectroscopy (NOESY) and Carr-Purcell-Meiboom-Gill (CPMG) pulse sequences. Macromolecular signals were removed with the NOESY presat (for water suppression) and the CPMG presat (as a T2 filter). Metabolite profiles were identified and analyzed using the Chenomx NMR Suite 8.4 professional software (Chenomx Inc., Edmonton, AB, Canada). The frequency position of tetramethylsilane (TMS) resonance was defined as exactly 0.0 ppm. The concentration of each metabolite was calculated after normalization for the area of the fixed TMS sample concentration added to the reaction. The raw data were analyzed using the build-in principal component analysis (PCA) and partial least squares discriminant analysis (PLS-DA) packages (available from the www.metaboanalyst.ca, accessed date 26 November 2020). We used the Variable Importance in Projection (VIP) values to express the contribution of each metabolite to the PLS-DA model. A *p* value < 0.05 and a VIP score > 1.1 were considered as significant.

### 4.6. Serum Biomarkers of Liver Damage

Blood samples were collected from experimental animals in tubes coated with lithium heparin (AmiShield, Taoyuan, Taiwan) on days 1 and 3 post-RT. Serum was isolated by centrifugation at 1500 rpm for 10 min. Quantification of AST, ALT, and albumin was carried out on a biochemical analyzer (AmiShield), according to the manufacturer’s protocol.

### 4.7. Immunohistochemical Staining of Inflammatory Markers

Mice were sacrificed at day 3 post-RT and tissues were embedded in the optimal cutting temperature compound for cryosection. Immunohistochemical staining was performed as previously described [16]. CD68 and F4/80 antibodies (Bio-Rad, Hercules, CA, USA) were used as macrophage markers. The results of immunostaining were quantified with the Image-Pro Plus software (Media Cybernetics, Silver Spring, MD, USA).

### 4.8. RNA Extraction and Gene Expression Analysis

Mice were sacrificed at days 1 and 3 post-RT and liver tissue samples were stored in liquid nitrogen. Total RNA extracted from normal tissue of the right and left lobes of the liver was isolated using the TRIzol reagent and reverse transcribed to cDNA with the Omniscript reverse transcriptase kit (Qiagen, Hilden, Germany). Quantitative PCR reactions were carried out with the LightCycler^®^ 480 SYBR Green I Master reagent (Roche Diagnostics Corporation, Indianapolis, IN, USA), and subsequently analyzed using the CFX ConnectTM Real-Time PCR Detection System (Bio-Rad). The primer sequences are shown in Appendix A.

### 4.9. Statistical Analysis

Continuous data are expressed as means ± standard deviations (SDs). Groups were compared on hepatic SUVs values and data obtained from NMR metabolomics experiments using the non-parametric Mann-Whitney U test. As for qPCR data, changes in expression for each of the investigated genes were determined by calculating the ΔΔCt values and compared with unpaired Student’s *t*-tests. All analyses were carried out in GraphPad Prism 6 (GraphPad Inc., San Diego, CA, USA). Statistical significance was determined by a two-tailed *p* value < 0.05.

## 5. Conclusions

The results of the current study demonstrate that experimental irradiation of the liver parenchyma results in dynamic metabolic changes that were detectable by different techniques (18F-FDG PET/CT imaging, NMR metabolomics, and qPCR) as early as one day post-RT. These findings have important clinical implications concerning the use of 18F-FDG PET/CT for monitoring patients with HCC treated with radiotherapy. Specifically, the increased uptake of 18F-FDG in this clinical population may, at least in part, reflect an enhanced glycolysis as an expression of radiation-induced alterations in the hepatic parenchyma. We conclude that 18F-FDG PET/CT should be used with caution in this clinical setting as it can yield false-positive results and may lead to erroneous estimates of tumor margins.

## Figures and Tables

**Figure 1 molecules-26-02573-f001:**
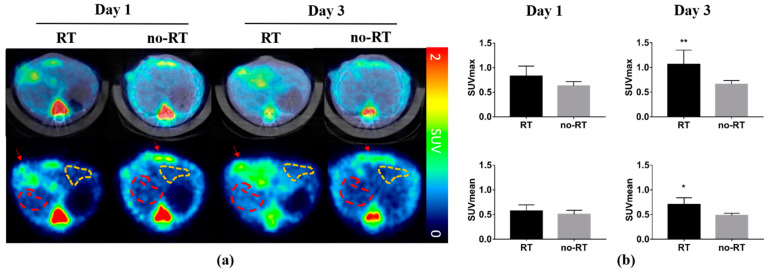
Impact of liver irradiation on 18F-FDG uptake by the areas surrounding orthotopic hepatocellular carcinomas in the right liver. (**a**) Representative transaxial images of fused PET/CT (upper row) and PET only (lower row) obtained in the RT (*n* = 5) and no-RT (*n* = 5) groups on the first and third post-irradiation days. Red and yellow dashed lines indicate the areas in the right and left liver lobes taken into account for the purpose of analysis. Red arrows indicate the experimental tumors. (**b**) SUVmean and SUVmax values measured in the right hepatic parenchyma on the first and third post-irradiation days. On the third post-irradiation day, both SUVmean and SUVmax values were significantly increased in the RT group, as compared with the no-RT group. * *p* < 0.05; ** *p* < 0.01.

**Figure 2 molecules-26-02573-f002:**
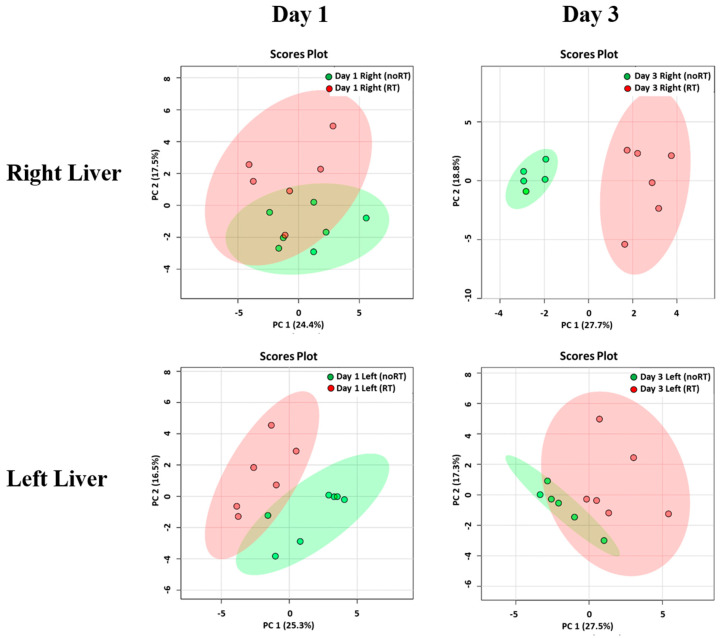
Principal component analysis of metabolite distribution in the RT (*n* = 6) and no-RT (*n* = 6) groups on the first and third post-irradiation days. Results are reported separately for the right and left lobes of the liver using partial least squares discriminant analysis. The areas (red for the RT group and green for the no-RT group) comprised within the elliptical rings indicate a 95% confidence level. A clear discrimination between the two experimental groups was observed for the right liver on the third post-irradiation day.

**Figure 3 molecules-26-02573-f003:**
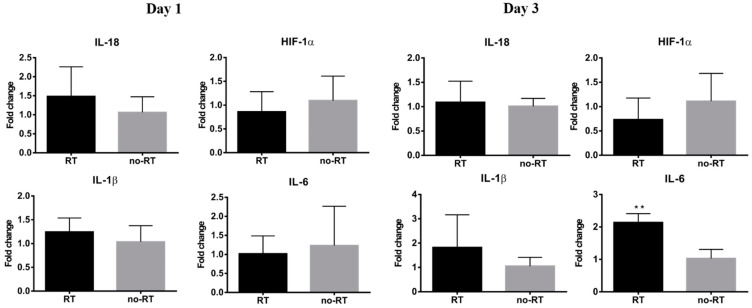
Expression levels of genes encoding for pro-inflammatory cytokines in the RT (*n* ≥ 3) and no-RT (*n* ≥ 3) groups on the first and third post-irradiation days (right lobe of the liver). On the third post-irradiation day, the expression levels of the IL-6 gene (expressed as fold-change) were significantly increased in the RT group—but not in the no-RT group. ** *p* < 0.01.

**Figure 4 molecules-26-02573-f004:**
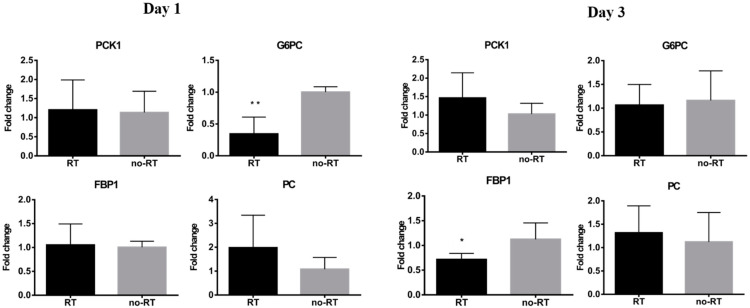
Expression levels of genes encoding for gluconeogenesis-associated enzymes in the RT (*n* ≥ 3) and no-RT (*n* ≥ 3) groups on the first and third post-irradiation days (right lobe of the liver). On the first post-irradiation day, the expression levels of the G6PC gene (expressed as fold-change) were significantly lower in the RT group. On the third post-irradiation day, the expression levels of the FBP1 gene (expressed as fold-change) were significantly lower in the RT group. These results indicate the temporal shift from gluconeogenesis to glycolysis in the irradiated liver parenchyma. * *p* < 0.05; ** *p* < 0.01.

**Figure 5 molecules-26-02573-f005:**
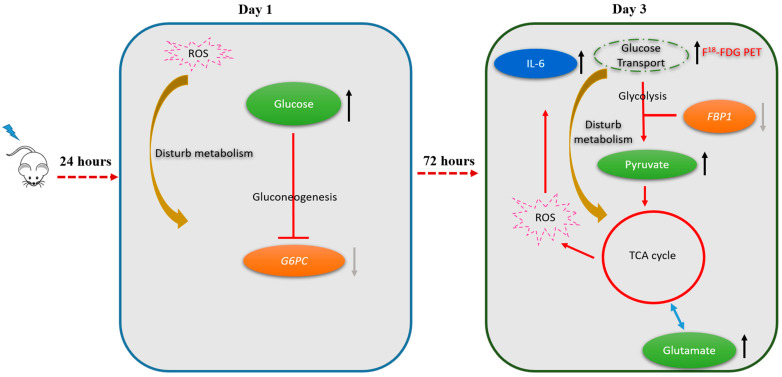
Shift in metabolic pathways from the first to the third post-irradiation day (right lobe of the liver). On the first post-irradiation day, radiation-induced production of reactive oxygen species (ROS) inhibited gluconeogenesis (as reflected by a reduced expression of the G6PC gene). On the third post-irradiation day, there was a shift towards glycolysis as reflected by a reduced expression of the FBP1 gene accompanied by an accumulation of pyruvate, glutamate, and the activation of the tricarboxylic acid cycle (TCA). The complex interplay between the activation of glycolysis and the presence of ROS also elicited an inflammatory response with an increased expression of IL-6. The metabolic shift occurring on day 3 can explain the significantly increased hepatic 18F-FDG uptake observed on 18F-FDG PET imaging at this time point.

**Table 1 molecules-26-02573-t001:** Dynamic changes in the expression of different metabolites in the right and left lobes of the liver observed in the two study groups on the first and third post-irradiation days.

Metabolite	VIP Score	Fold Change (|RT/no-RT| > 1.1)	*p*	Metabolite	VIP Score	Fold change (|RT/no-RT| > 1.1)	*p*
*Right liver*		*Day 1*				*Day 3*	
Glucose	3.456	16.930	0.003	Pyruvate	1.818	6.996	0.008
Sucrose	2.244	13.483	0.015	Glutamate	1.043	3.487	0.024
Galactarate	1.396	2.523	0.003	Sucrose	1.153	2.812	0.019
Galactitol	1.931	4.776	0.001	Malonate	0.879	0.565	1.6 × 10^−4^
				Pyridoxine	0.836	0.512	0.006
				Choline	0.955	0.479	0.009
				Niacinamide	1.244	0.345	1.7 × 10^−7^
				Hypoxanthine	1.763	0.341	0.028
				Betaine	1.449	0.252	1.4 × 10^−4^
				Guanidoacetate	2.690	0.121	0.001
				Sarcosine	1.992	0.090	3.8 × 10^−4^
				Glycocholate	2.190	0.059	2.0 × 10^−4^
*Left liver*		*Day 1*				*Day 3*	
Glucose	2.561	17.774	0.008	Fumarate	1.108	1.844	0.001
Malonate	2.597	9.836	0.001	Niacinamide	0.981	0.577	0.005
Succinate	2.437	3.979	0.004	Riboflavin	1.062	0.452	0.029
Galactitol	1.539	3.506	0.001	Succinate	1.200	0.428	0.008
Glycylproline	1.350	3.018	1.7 × 10^−4^	Succinylacetone	1.326	0.415	0.043
Galactarate	1.404	2.866	0.001	Betaine	1.492	0.364	3.2 × 10^−4^
Alanine	1.185	2.093	0.018	Sarcosine	1.448	0.349	0.002
N-Methylhydantoin	0.919	1.832	0.049	Guanidoacetate	2.401	0.335	0.036
Anserine	0.613	1.433	0.046				

**Table 2 molecules-26-02573-t002:** Number of animals used for each procedure at different time points during the experimental workflow.

Experimental Design		Number of Animals			
Post-irradiaton				1 day	3 day
Longitudinal imaging data	FDG PET		^1^RT	5	5
			no-RT	5	5
Time-points data collection	NMR metabolomics		^1^RT	6	6
			no-RT	6	6
	Biomarkers of liver damage		^1^RT	3	3
			no-RT	3	3
	Cytokines qPCR		^1^RT	≥3	≥3
			no-RT	≥3	≥3
	Gluconeogenesis qPCR		^1^RT	≥3	≥3
			no-RT	≥3	≥3

^1^ The right lobe of each mouse in the RT group was received 15 Gy irradiation.

## Data Availability

The datasets used and/or analyzed during the current study are available from the corresponding author on reasonable request.

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
