# Peer review of "Radiation-Induced Metabolic Shifts in the Hepatic Parenchyma: Findings from 18F-FDG PET Imaging and Tissue NMR Metabolomics in a Mouse Model for Hepatocellular Carcinoma"

_molecules, 2021, doi:10.3390/molecules26092573_

Round 1

Reviewer 1 Report

Yi-Hsiu Chung et al reported metabolic shifts in the hepatic parenchyma in a mouse model for hepatocellular carcinoma by radiation by using 18F-FDG PET imaging and tissue NMR metabolomics. The results are well interpreted and presented. I only have a few comments.

  1. The number of animals used in each group should be presented figure legends.
  2. In the legend of Figure 1, it is confusing that “both SUVmean and SUVmax values were significantly increased in the RT group – but not in the no-RT group.” Is that means these values increased in RT group compared to no-RT group (control group)?
  3. Are the liver tissues (right versus left, tumor versus non-tumor) analyzed separately? The authors are suggested to discuss the rationale of metabolites of choosing the area of tissue for metabolomics analysis.
  4. The authors are suggested to describe the detailed method for metabolomics analysis, such as the level of metabolite identification, are these observation were confirmed by authentic standard? Does the authors look into lipid alterations in these models?
  5. The authors are suggested to discuss the effects of radiation on phenotype of tumor HCC development, and its relation to metabolome alterations.

Reviewer 2 Report

The authors present a number of interesting observations regarding metabolic shifts observed after irradiation of the right hand lobes of mouse livers  implanted with hepatocellular carcinoma cells (BNL cells).  Overall a decrease in gluconeogenesis and an increase in glycolysis is noted by 3 days post-irradiation and the authors speculate that increased 18F-FDG uptake in the right tumor bearing side of mice irradiated specifically on the right liver as compared to the right tumor bearing side of livers from non-irradiated mice may be due to this shift to glycolysis. 

Although a number of interesting studies are conducted, there are some clarifications needed

1) The authors spend a lot of time in the introduction discussing the pathology associated with radiation induced liver damage (RILD) and they seem to intimate that the metabolic shifts observed are in some way tied to this pathology.  There is no evidence to directly connect them.  Unless some more direct evidence is presented, the authors should temper their discussion of this connection.

2) The authors present 18F-FDG data collected from a cohort of mice used in a previous study (published in 2019) together with metabolomic data collected from a cohort of animals used here.  There should be some discussion of how closely related these cohorts of animals are to ensure that the different sets of data can be fairly compared and the events noted in each can be assumed to be occurring simultaneously - i.e. there should be a description in the methods of the vendor, age, gender, strain, housing conditions, etc of the mice used for both sets of data collection.

3) Please show error bars for no-RT data in Figures 3 and 4.

4) Please move figure legend to be next to Figure 3.

5) The concluding statement ". We conclude that 18F-FDG PET/CT should be used with caution in this clinical setting as it can yield false-positive results and may lead to erroneous estimates of tumor margins." seems somewhat out of place.  There has been no previous discussion of the effect of 18F-FDG data on tumor margins.  Some additional discussion on this point should be mentioned prior to the conclusion.

Round 2

Reviewer 1 Report

The authors have addressed most of the questions. The manuscript is ready to be published.